# A Sensorless Control Strategy for Permanent Magnet Synchronous Motor at Low Switching Frequency †

**Zhao Xue \*, Lin Li, Xiaolu Wang and Xin Wang**

Beijing Institute of Precision Mechatronics and Controls, Beijing 100076, China; bipmc_lil@outlook.com (L.L.); wxlu0517@gmail.com (X.W.); bipmc_wangx@outlook.com (X.W.)

\* Correspondence: xuezh_bipmc@outlook.com

† This paper is an extended version of our paper published in the 23rd International Conference of Fluid Power and Mechatronic Control Engineering (ICFPMCE) under the title "A Sensorless Control Strategy for Permanent Magnet Synchro-nous Motor at Low Switching Frequency". 22–24 July 2022, Kunming, China.

**Abstract:** The high-frequency (HF) square-wave voltage injection method can be used in permanent magnet synchronous motor (PMSM) drive systems. However, when the switching frequency is too low, the injection frequency will also decrease, which will reduce the update frequency of the HF response current, making it difficult to extract the position quadrature signal and affecting the accuracy of position estimation and control performance. This paper proposes a method for extracting position quadrature signals based on sampling rate transformation, and a signal processing strategy based on Cascade Integrator Comb (CIC) interpolation filtering, which can solve the problem of waveform distortion caused by the low sampling rate of the extracted position quadrature signal. This strategy can increase the sampling rate of the position quadrature signal to the pulse width modulation (PWM) update frequency by interpolating in the sampling current, thereby reducing the harmonic content of the position quadrature signal and improving the position estimation. precision. In addition, the PWM update frequency and estimated rotational speed information are used to compensate for the delay caused by position estimation and inverter update, which effectively improves the accuracy of position estimation. Finally, the effectiveness of the proposed control strategy is verified by simulation.

**Keywords:** permanent magnet synchronous motor; sensorless control; cascade integrator comb interpolation filtering; delay compensation



## 1. Introduction

Compared with the induction motor, permanent magnet synchronous motor (PMSM) has higher efficiency and higher power density, so it is widely used in high-performance applications [1,2]. In the electric traction system of rail transit, due to the harsh operating environment and the small installation position of the sensor, it is not suitable to use the position sensor to realize the closed-loop control based on PMSM. The sensorless control strategy can solve this problem [3]. Sensorless control strategies are mainly divided into two categories. One is model-based sensorless control suitable for medium and high-speed areas, and the other is sensorless control based on salient polarity suitable for zero and low speed areas [4]. As a position estimation method based on salient polarity, the high-frequency (HF) injection method can generally be divided into current injection and voltage injection according to different injection forms. Since the estimation bandwidth of rotor position is affected by the bandwidth of current controller, the application of current injection method is relatively few [5]. HF voltage injection method can be divided into indirect flux detection by online reactance measurement (INFROM) method [6], rotating sinusoidal injection method [7], pulse vibration sinusoidal injection method [8,9], pulse vibration square wave injection method [10], and pulse signal injection method [11]. The difference lies in the form and time of injection signal and the demodulation method of HF

response current. In this paper, the HF voltage injection method based on saliency polarity is adopted, which can ensure a high signal-to-noise ratio and improve the stability of the control system under harsh working conditions.

In addition, for the electric traction system, in order to improve the efficiency and load-carrying capacity of the inverter and avoid excessive switching loss, the switching frequency will generally not exceed 1 kHz [12], which will limit the maximum injection frequency of the HF voltage injection method and reduce the update frequency of the estimated position, so that the rotor position estimation error increases in the range of zero and low speed, the current harmonic content increases, and the HF noise is more obvious. In order to reduce current harmonics and HF noise, the method of reducing the injection frequency avoids the more sensitive frequency range of human ears [13]. However, this method makes the frequencies of the fundamental frequency signal and the high-frequency response signal closer, which further limits the low-speed operating range and dynamic performance, and causes the neglected resistance term in the high-frequency model to affect the position estimation accuracy [14–16]. In addition, reducing the amplitude of the injected signal can reduce the noise amplitude [17], but it will reduce the signal-to-noise ratio and affect the accuracy of position estimation. Although the pseudo-random HF square wave voltage injection method [18], the dual-frequency random HF voltage injection method [19], and the intermittent pulse injection method [20] can reduce the noise, they all need a higher switching frequency, so that one or more injected HF signals can be inserted in multiple carrier cycles, which is not suitable for the case of low switching frequency.

In order to solve these problems and improve the position observation accuracy and control performance at low switching frequency, this paper, based on the high-frequency square wave voltage injection method, proposes a method for extracting position quadrature signals based on sampling rate transformation to avoid the use of complex modulation methods, and adopts the signal processing strategy based on the CIC interpolation filtering to solve the problem of mismatch between the update frequency of the position quadrature signal and the PWM update frequency, and suppress the HF ripple caused by the low update frequency of the position quadrature signal. At the same time, the delay compensation strategy is used to compensate for the delay caused by position estimation and inverter update, which effectively improves the accuracy of position estimation. Finally, the performance of the proposed strategy is analyzed by simulation.

## 2. Sensorless Control Strategy at Low Switching Frequency

The voltage equation of PMSM in *d-q* rotating coordinate system is [21,22]:

$$\begin{bmatrix} u_d \\ u_q \end{bmatrix} = \begin{bmatrix} R_s + L_d p & -\omega_e L_q \\ \omega_e L_d & R_s + L_q p \end{bmatrix} \begin{bmatrix} i_d \\ i_q \end{bmatrix} + \begin{bmatrix} 0 \\ \omega_e \psi_f \end{bmatrix} \tag{1}$$

where $u_{dq}$, $i_{dq}$ and $L_{dq}$ are stator voltage, stator current and stator inductance in *d-q* rotating coordinate system respectively; $R_s$ is the stator resistance; $\omega_e$ is the electrical angular velocity; $p$ is differential operator, $p = d/dt$; $\Psi_f$ is the flux linkage.

In the HF-response model, because the injected square wave voltage signal frequency is higher than the motor operating frequency, the induced voltage drop of the inductance in the motor winding is greater than that of the resistance. In order to facilitate subsequent analysis and calculation, the voltage drop on the stator, the voltage drop of the inductance and the back EMF can be ignored. The HF mathematical model of PMSM can be simplified as follows:

$$\begin{bmatrix} u_{sd} \\ u_{sq} \end{bmatrix} = \begin{bmatrix} pL_d & 0 \\ 0 & pL_q \end{bmatrix} \begin{bmatrix} i_{dh} \\ i_{qh} \end{bmatrix} \tag{2}$$

where $u_{sd}$ and $u_{sq}$ are the HF voltage components in the *d-q* rotating coordinate system respectively; $i_{dh}$ and $i_{qh}$ are the HF response current components in *d-q* rotating coordinate system respectively.

The square wave voltage injection method is to inject a square wave voltage signal into the estimated *d*-axis, so that the motor winding can excite HF response current containing rotor position information. Generally, the injection method can be divided into discontinuous pulse injection form and continuous pulse injection form according to the control cycle and whether the injection cycle is continuous. The schematic diagram is shown in Figure 1.

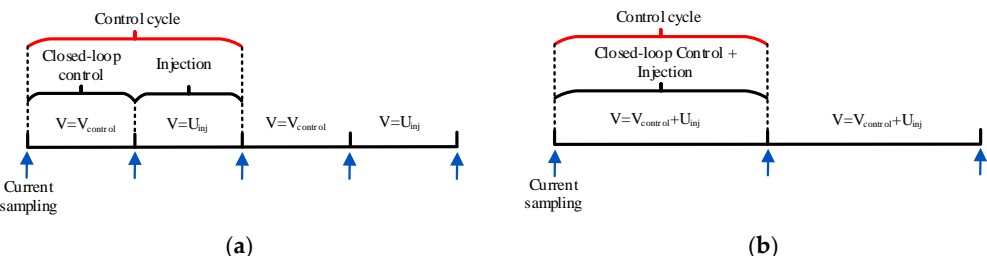

**Figure 1.** Different voltage injection forms. (**a**) Schematic diagram of discontinuous pulse injection; (**b**) Schematic diagram of continuous pulse injection.

The discontinuous pulse injection strategy realizes position estimation in the injection cycle and closed-loop control in the control cycle. This method can reduce the injected frequency and noise, reduce the influence of harmonics on position estimation, improve the signal-to-noise ratio, and improve the accuracy of position solution without adding filters [18].

However, the discontinuous pulse injection strategy needs to make the switching frequency much greater than the operating frequency, so as to reduce the excessive current disturbance caused by switching and inserting a discontinuous injection HF signal in multiple control cycles. At low switching frequency, discontinuous signal injection increases the interval of injected waveform, resulting in the corresponding increase of the interval of extracted position information. Excessive delay is unacceptable to the system.

Therefore, the continuous injection mode is adopted in this paper, that is, the HF voltage pulse signal with positive and negative opposite directions is injected in each carrier cycle. As shown in Equation (3), it is the expression of square wave voltage injected in the d-axis of the estimated rotating coordinate system:

$$u_{\hat{d}q} = \begin{cases} U_h, t \in [0, \frac{T_h}{2}] \\ -U_h, t \in [\frac{T_h}{2}, T_h] \end{cases} \tag{3}$$

where $U_h$ is the amplitude of the injected square wave voltage; $T_h$ is the injection period of the HF square wave voltage, $T_h$ is equal to half the control period. Transforming the injected HF voltage signal into the *d-q* rotating coordinate system, there are:

$$u_{dq} = u_{\hat{d}q} e^{-j(\theta_e - \hat{\theta}_e)} = u_{\hat{d}q} e^{-j\Delta\theta} \tag{4}$$

where $\Delta\theta = \theta_e - \hat{\theta}_e$. $\theta_e$ is the actual electrical angle and $\hat{\theta}_e$ is the estimated electrical angle. The resulting HF response current in the *α-β* stationary coordinate system is expressed as:

$$i_{\alpha\beta} = \left(\frac{\cos\Delta\theta}{L_d}\int u_{\hat{d}q}dt - j\frac{\sin\Delta\theta}{L_q}\int u_{\hat{d}q}dt\right) \times (\cos\theta_e + j\sin\theta_e) \tag{5}$$

By extracting the envelope, the Equation (6) can be obtained:

$$\begin{bmatrix} \Delta i_{\alpha h} \\ \Delta i_{\beta h} \end{bmatrix} = \frac{\Delta U}{L_d L_q \omega_h} \begin{bmatrix} L_{avg}\cos\hat{\theta}_e - L_{dif}\cos(2\theta_e - \hat{\theta}_e) \\ L_{avg}\sin\hat{\theta}_e - L_{dif}\sin(2\theta_e - \hat{\theta}_e) \end{bmatrix} \tag{6}$$

where $\Delta i_{\alpha h}$ and $\Delta i_{\beta h}$ represent the envelopes of the two HF response currents in the stationary coordinate system, respectively; $L_{avg} = (L_d + L_q)/2$, $L_{dif} = (L_d - L_q)/2$, $\Delta U$ represents

the amplitude. The magnitude is $U_h$, and the direction of the voltage changes with the periodic positive and negative changes of the injected square wave; $\omega_h$ is the frequency of the injected HF signal. When the estimated position error is approximately zero, the envelope $\Delta i_{\alpha h}$ and $\Delta i_{\beta h}$ approximation have a positional quadrature relationship.

It can be seen from the above formula that the rotor position is included in the HF response current envelope $\theta_e$. Therefore, the HF response current can be extracted from the feedback current, the rotor position information can be obtained by using the appropriate demodulation method, and the rotor position and mechanical angular velocity can be estimated by the observer, so as to finally realize the closed-loop control.

As shown in Figure 2, it is the block diagram of the PMSM control system using the square wave voltage injection method proposed in this paper, mainly including the rate conversion to extract the envelope current, and the cascade integrator comb (CIC) interpolation filtering to reduces the harmonic content of the signal, the phase-locked loop (PLL) and the Luenberger observer to estimate rotor position and speed, et al. In addition, the closed-loop control strategy adopts field-oriented control (FOC) based on proportional-integral (PI) controller. The principle is described in subsequent sections.

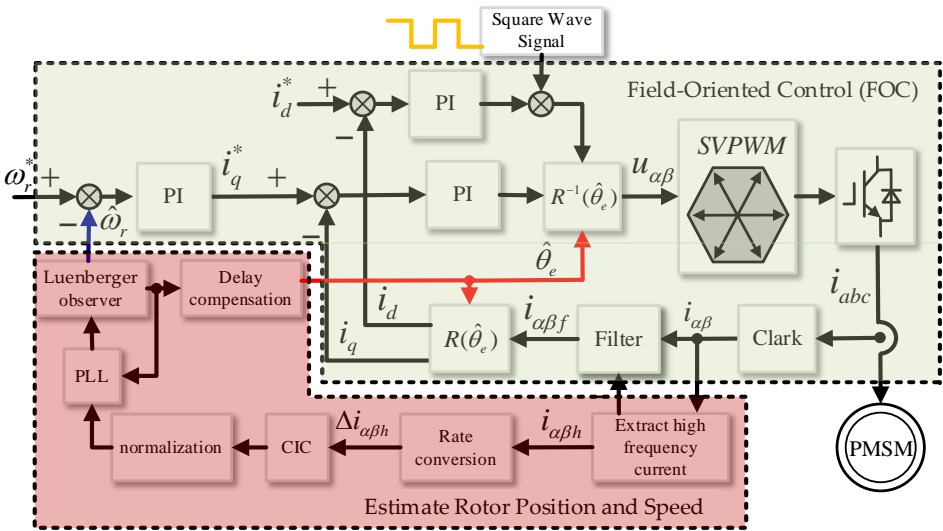

**Figure 2.** Block diagram of PMSM control principle based on square-wave voltage injection method.

## 3. Rotor Position Estimation Method

*3.1. High Frequency Response Current Envelope Extraction Method*

Feedback current $i_{\alpha\beta}$ including fundamental current $i_{\alpha\beta s}$ and HF response current $i_{\alpha\beta h}$. Therefore, it is necessary to extract the HF response current from the feedback current, that is, to realize the demodulation of the HF signal and fundamental signal. The usual method is to use a high pass filter [23], but there will be serious phase delay, and the filtering effect is poor when the injection frequency is close to the fundamental frequency. In addition, dual frequency notch filter [24], second-order generalized integral [25], and other methods are also used to extract HF response current, but the implementation process is more complex.

In order to solve the above problems and simplify the process of high-frequency signal extraction, this paper adopts a HF response current extraction strategy based on the delay module. Since the frequency of the injected HF voltage signal is much higher than the operating frequency, the fundamental current $i_f(k)$ at time $k$ can be approximately equal to the fundamental current $i_f(k + T_h/2)$ at time k. Since the signs of the injected HF voltage signals at time $k$ and $(k + T_h/2)$ are opposite, approximately $i_{\alpha\beta h}(k) = -i_{\alpha\beta h}(k + T_h/2)$. Therefore, the HF response current is obtained by making a difference between the feedback currents at different sampling times. The equation is as follows:

$$i_{\alpha\beta h}(k) = (i_{\alpha\beta}(k) - i_{\alpha\beta}(k + T_h/2))/2 \tag{7}$$

The above method realizes the approximate HF response current extraction without filter. The method does not produce a large phase delay for the extracted HF response signal, and is not affected by switching frequency and injection frequency. The sampling rate of the HF response current obtained at this time is $f_s = 1/(T_h/2)$, and the schematic diagram is shown below.

In Figure 3, the transformation multiple of sampling rate $R = T_{PWM}/(T_h/2)$, which is used to measure the change of rate. When $R < 1$, it means that the sampling rate decreases, and when $R > 1$, it means that the sampling rate increases; $T_{PWM}$ is the switching cycle; Th is the square wave voltage injection period.

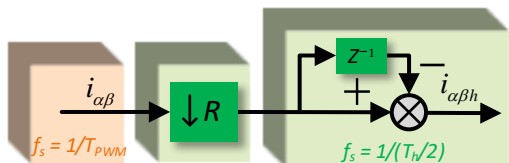

**Figure 3.** Schematic diagram of HF-response current extraction.

Next, the envelope of the H- response current, that is, the position quadrature signal, is extracted to estimate the rotor position information. In article [26], QR decomposition is used to extract HF current envelope to reduce the phase lag of the demodulation link, which is relatively complex. Article [27] proposed an envelope extractor. This method only needs the delay model and the symbol of the injected signal, and the HF current envelope can be obtained through subtraction.

However, this method needs to select the sampling point at the midpoint of the period of the injected HF square wave voltage signal, and consider the influence of the delay of the digital control system on the symbol judgment of the injected signal and the influence of the sampling point. At low switching frequency, the method also needs to put a low-pass filter in series at the output to eliminate the disturbance in the extracted quadrature signal, resulting in the increase of control delay.

In order to solve this problem, and to conveniently and effectively extract the position quadrature signal at low switching frequency, this section based on the sampling holding function in the process of sampling rate transformation, and the corresponding sampling rate is $f_s = 1/T_h$. The schematic diagram of the sampling rate conversion process is as follows.

In Figure 4, the conversion multiple of sampling rate $R = 0.5$, Th is the square wave voltage injection period. The method has a simple structure, does not need the symbol information of the injected square wave voltage, and is not affected by the change of switching period. As shown in the Figure 5, when the injected square wave voltage frequency is 500 Hz and the motor is in motion, the position quadrature signal can be extracted by using the sampling rate conversion process.

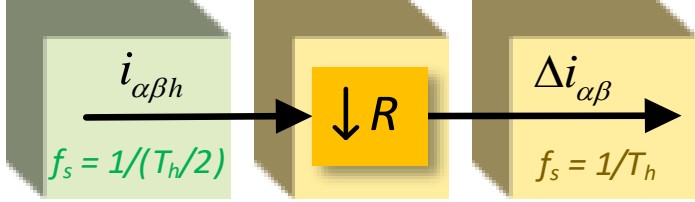

**Figure 4.** Schematic diagram of sampling rate conversion process.

It can be seen from the Figure 5 that although the position quadrature signal can be accurately extracted based on the sampling rate transformation process, due to the low injection frequency, the obtained position quadrature signal changes in steps and has serious distortion, which will lead to obvious steps in the subsequent position estimation and eventually affect the stability of the system. In addition, when the selected square

wave voltage injection frequency is reduced, the discretization of the extracted position quadrature signal will be more serious. As shown in Figure 6, it is the estimated rotor position calculated by directly using the position quadrature signal.

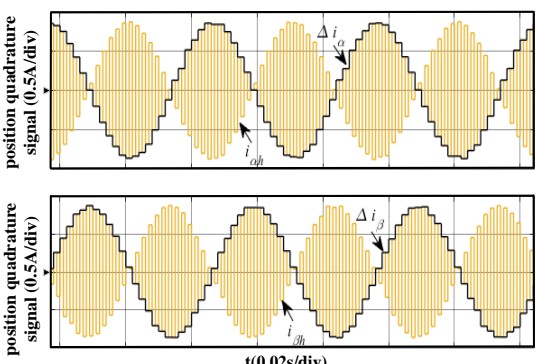

**Figure 5.** Schematic diagram of position quadrature signal extraction.

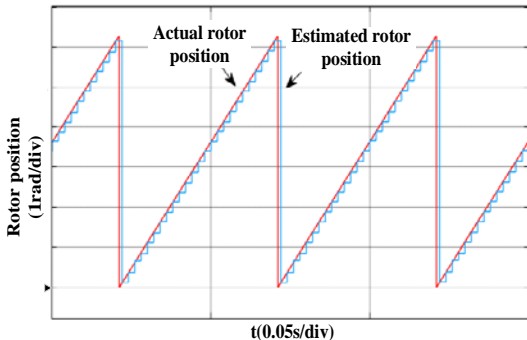

**Figure 6.** Estimation of rotor position using position quadrature signal calculation.

### 3.2. Frequency Matching Method Based on CIC Interpolation Filtering

A filtering method based on quadratic interpolation is proposed in [28], which improves the continuity of the position quadrature signal after quadratic interpolation filtering and effectively eliminates the harmonic components related to the injected signal.

However, this method uses the current sampling value to calculate the fitting value at the last sampling time, so a phase lag will occur, and the lag angle is proportional to the sampling period, so the lower the injection frequency, the larger the phase lag. In addition, the difference between the sampling frequency and the switching frequency of the position quadrature signal is too large, that is, when the switching frequency is more than twice the sampling frequency, the quadratic interpolation can only double the sampling points, which still cannot effectively solve the waveform distortion caused by the sampling holding effect.

In order to solve the above problems, in this section, a frequency-matching method based on CIC interpolation filtering is proposed to interpolate and to match the frequencies of the discrete signals, so that the interpolated quadrature signal has an update frequency consistent with the switching frequency, and the waveform is more continuous and complete. As shown in Figure 7, it is the block diagram of single-order CIC interpolation filter. As can be seen from Figure 7, the filter is composed of a comb filter $H_1(z)$ and integrator $H_2(z)$ in cascade. It only contains addition and subtraction modules, so it has the advantages of simple structure and low complexity [29]. In addition, the CIC filter is a multi-rate low-pass filter, which is suitable for sampling rate conversion. It can not only realize the extraction from high rate to a low rate, but also realize the interpolation from low rate to high rate.

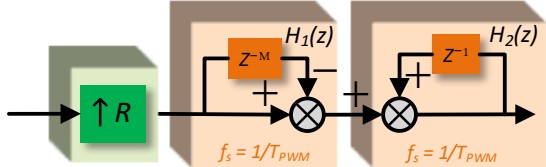

**Figure 7.** Block diagram of single order CIC filter.

In position observation, CIC filter can be used to interpolate the position quadrature signal with low sampling rate. The interpolation points increase by the same factor as the sampling rate. Finally, the quadrature signal with sampling rate consistent with the switching frequency can be obtained, the zero-order holder phenomenon caused by low sampling rate can be improved, and the calculation period and current harmonic can be reduced. where $M$ is the order of the filter, that is, the number of beats delayed, which determines the change speed of the output response in the turning region; The transformation multiple of the sampling rate $R = M$, which determines the number of sampling points inserted by the output signal in the adjacent time interval of the input signal. The system function $H(z)$ of single-order CIC filter can be expressed as:

$$
\begin{aligned}
H(z) &= H_1(z) \times H_2(z) \\
&= \left(1 - z^{-M}\right) \times \left(\frac{1}{1-z^{-1}}\right) \\
&= \frac{1-z^{-M}}{1-z^{-1}}
\end{aligned}
\tag{8}
$$

According to the Noble identity transformation formula, the single-order CIC filter can be equivalently transformed into the following simpler form:

In Figure 8, $R = (T_h/2)/T_{PWM}$, that is, the sampling rate of the position quadrature signal is increased to the switching frequency through interpolation. The multi-order CIC filter is realized by cascading multiple single-order CIC filters to meet the larger amplitude attenuation requirements, but the phase delay will also increase. In this paper, the single-order CIC filter is used. The equivalent structure of the filter can meet the needs of interpolation filtering. As shown in Figure 9, it is a schematic diagram of the interpolation effect after the sampling rate is doubled.

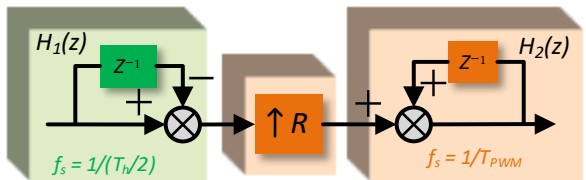

**Figure 8.** Equivalent block diagram of a single-order CIC filter.

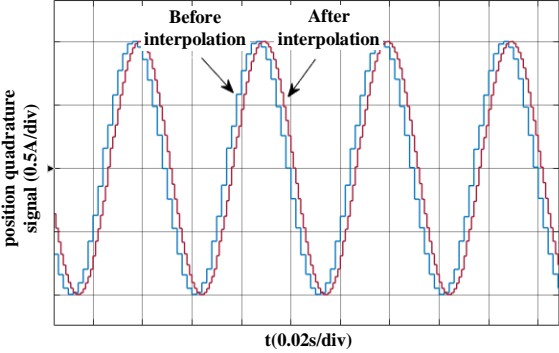

**Figure 9.** Schematic diagram of single-order CIC interpolation filtering.

### 3.3. Observer Design and Stability Analysis

After normalizing the interpolated position quadrature signal, the rotor position error is obtained through the quadrature phase-locked loop, and then the estimated rotor position $\hat{\theta}_e$ and mechanical angular velocity $\hat{\omega}_r$ are calculated by the Luenberger observer. The principle of the Luenberger observer is detailed in [30,31]. The general control delay varies from $0.75T_{PWM}$ to $2T_{PWM}$ depending on the current update-sampling processing method. Considering the position observation, the maximum delay in the zero-low speed range does not exceed $1.5T_{PWM}$. Therefore, dynamic angle compensation is performed on the estimated rotor position, and the compensation amount is selected as $\Delta\theta = 3.5T_{PWM}\hat{\omega}_e$, $\hat{\omega}_e$ is the estimated electrical angle. As shown in Figure 10, it is a block diagram of the position observation principle based on CIC filtering.

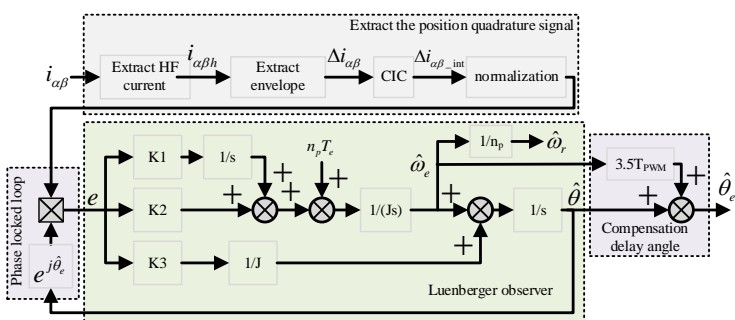

**Figure 10.** Block diagram of position observation based on CIC filtering.

The open-loop transfer function of the Luenberger observer is shown in the following equation. Through this observer, the rotor position and speed of the motor can be estimated.

$$\begin{cases} \hat{\theta} = \left( \frac{K_1 + sK_2 + sn_pT_e}{s^3 J} + \frac{K_3}{sJ} \right) \times e \\ \hat{\omega}_e = \left( \frac{K_1 + sK_2 + sn_pT_e}{Js^2} \right) \times e \end{cases} \tag{9}$$

where $e$ is the input to this observer, and represents position estimation error; s represents complex variable; $T_e$ and $n_p$ represent the electromagnetic torque and number of pole pairs, respectively; $\hat{\theta}$ and $\hat{\omega}_e$ are estimated rotor position and estimated electrical angle speed, respectively; $K_1$, $K_2$ and $K_3$ are the variable parameters of the observer, respectively. $J$ represents the moment of inertia.

In order to analyze the stability of the Luenberger observer, and the robustness of parameter changes, the phase locked loop needs to be equivalent to the actual rotor position $\theta_e$ minus the estimated rotor position $\hat{\theta}$ in Figure 10, that is, $e = \theta_e - \hat{\theta}$. Since the compensated delay angle does not affect the transfer function of the observer, that is, it does not affect its stability, so it can be ignored. Then the transfer function can be obtained from Figure 10 as follows:

$$\hat{\theta} = \frac{sT_e n_p}{s^3 J + s^2 K_3 + sK_2 + K_1} + \frac{(s^2 K_3 + sK_2 + K_1) \times \theta_e}{s^3 J + s^2 K_3 + sK_2 + K_1} \tag{10}$$

where s represents complex variable; $T_e$ and $n_p$ represent the electromagnetic torque and number of pole pairs of PMSM, respectively; $\theta_e$ and $\hat{\theta}$ represent actual rotor position and estimated rotor position, respectively; $K_1$, $K_2$ and $K_3$ are the variable parameters of the observer, respectively. $J$ represents the moment of inertia. The motion equation of the motor is:

$$\theta_e = \frac{(T_e - T_L) \times n_p}{s^2 J} \tag{11}$$

Combining Equations (10) and (11), the following can be obtained:

$$e = \frac{-s n_p T_L}{s^3 J + s^2 K_3 + s K_2 + K_1} \tag{12}$$

According to the final value theorem, when the load torque is a step signal or a ramp signal, the steady-state error of the observer is 0, so the system has better dynamic performance. When considering observer stability, the closed-loop pole needs to be located on the negative real axis of the s-plane. The general practice is to equal the three poles and make the eigenvalues greater than zero, that is, $s_1 = s_2 = s_3 = a$ ($a > 0$). The value of $a$ should be adjusted based on the actual situation. Therefore, the theoretical observer parameters are $K_1 = a^3 J$, $K_2 = 3a^2 J$, $K_3 = 3aJ$, respectively.

### 4. Simulation Analysis

In order to verify the effectiveness of the strategy proposed in this paper, this section will be verified by simulation. The parameters of the PMSM used in the simulation are shown in Table 1.

**Table 1.** Permanent magnet synchronous motor parameters.

| Parameter | Value [Unit] |
|---|---|
| Rated power | 3 [kW] |
| Rated voltage | 540 [VDC] |
| Rated speed | 1200 [rpm] |
| Stator resistance | 0.19 [Ω] |
| *d*-axis inductance | 3.53 [mH] |
| *q*-axis inductance | 7.48 [mH] |
| Number of pole pairs | 4 |

The position quadrature signal extracted from the HF corresponding current is tested by simulation. The simulation condition is set as a no-load condition, the reference speed is 300 rpm, the frequency of injected square wave voltage is 500 Hz, and the switching frequency is 1 kHz.

Figure 11 shows the position quadrature signal without CIC interpolation filtering and its FFT analysis diagram, and Figure 12 shows the position quadrature signal with CIC interpolation filtering and its FFT analysis diagram. By comparing Figures 11a and 12a, it can be seen that the position quadrature signal without CIC interpolation filtering has obvious discontinuity, and its harmonic content is richer, which has an adverse impact on the subsequent position estimation and increases the estimated position error. By comparing Figures 11b and 12b, it can be seen that the waveform of position quadrature signal filtered by CIC interpolation is smoother, the harmonic content is significantly reduced, and the interference of HF signal can be well filtered. In addition, after interpolation, the sampling rate of the waveform is increased to the switching frequency, which can improve the calculation frequency of the estimated position, and improve the accuracy of position estimation.

Then, the steady-state performance of the proposed control strategy is tested by simulation. Figures 13 and 14 are waveforms of the estimated position signal reference 100 rpm and 300 rpm under no-load conditions, respectively. It can be seen from the figure that the maximum position estimation error is no more than 0.1 rad, the observer can better estimate the rotor position at different reference speeds. Therefore, the proposed strategy can be well applied to sensorless control in the zero low speed range at 1 kHz low switching frequency.

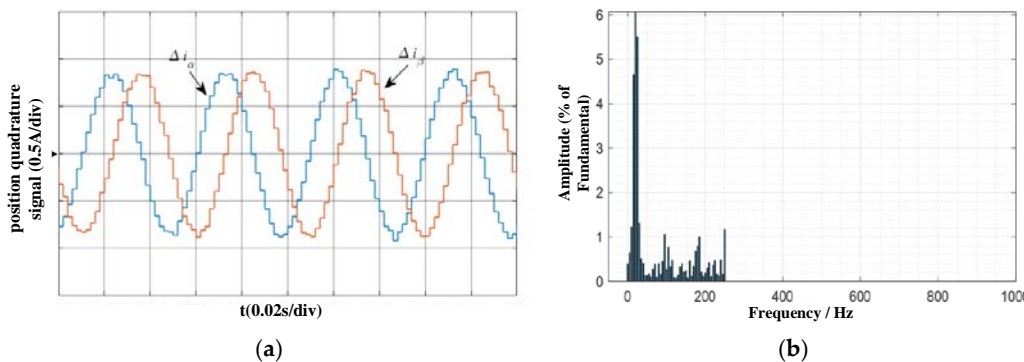

**Figure 11.** Position quadrature signal without CIC interpolation filter and its FFT analysis. (**a**) Position quadrature signal; (**b**) FFT analysis.

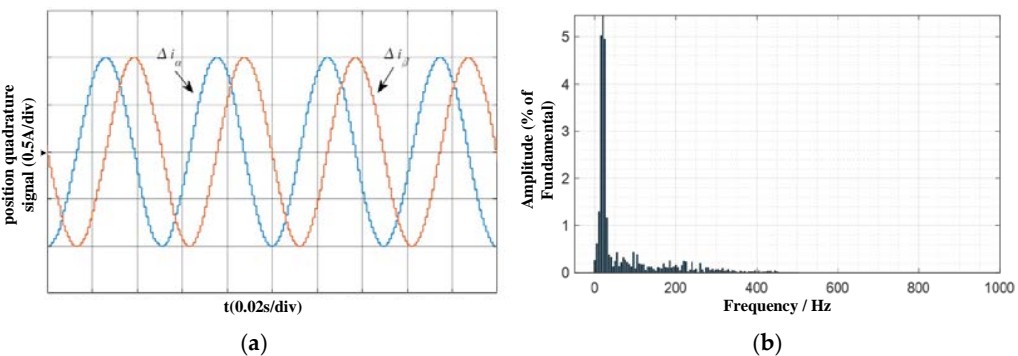

**Figure 12.** Position quadrature signal filtered by CIC interpolation and its FFT analysis. (**a**) Position quadrature signal; (**b**) FFT analysis.

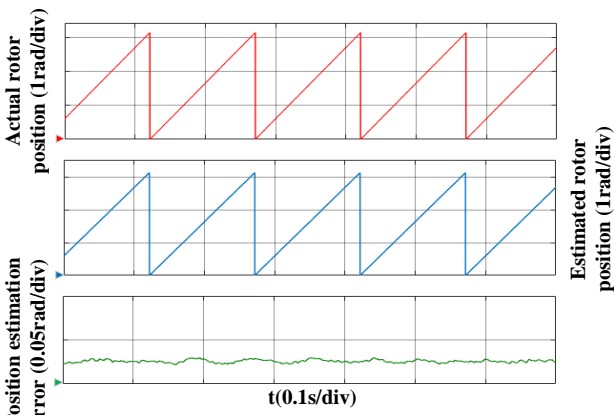

**Figure 13.** Simulation waveform at a reference speed of 100 rpm under no-load.

In addition, the dynamic characteristics under the proposed control strategy are tested by simulation. Figures 15 and 16 show the estimated position waveforms and the corresponding phase current waveforms at a reference 100 rpm and 300 rpm in the case of a sudden load of 3 N.m. It can be seen from the figure that in the whole loading process, the maximum position estimation error does not exceed 0.26 rad., Therefore, the control system has strong anti-disturbance under different reference speeds.

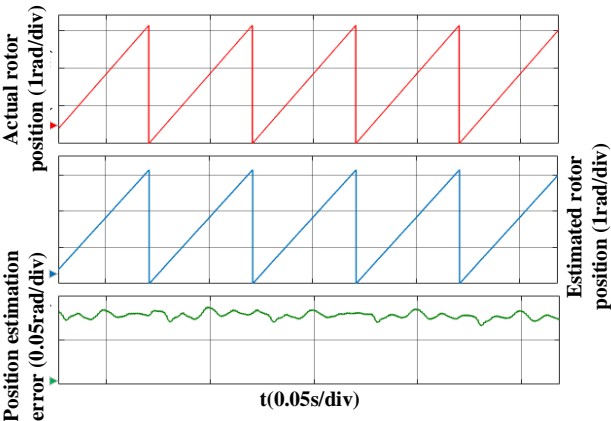

**Figure 14.** Simulation waveform at a reference speed of 300 rpm under no-load.

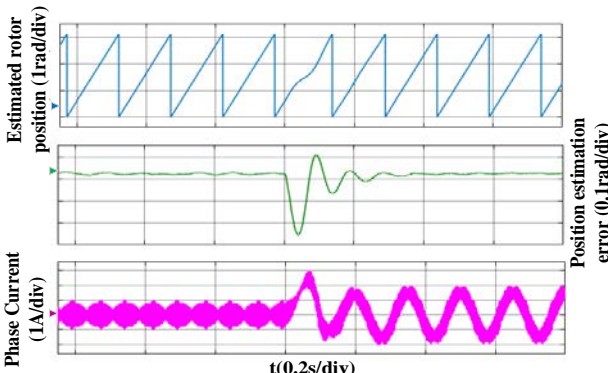

**Figure 15.** Sudden load simulation waveform at a reference speed of 100 rpm.

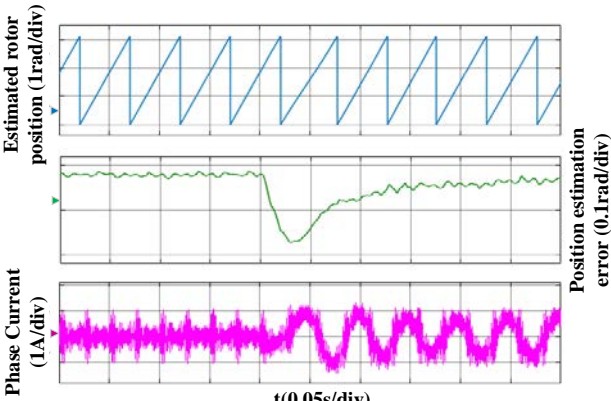

**Figure 16.** Sudden load simulation waveform at a reference speed of 300 rpm.

Finally, the starting and dynamic characteristics of the sensorless control system is tested by simulation. Figure 17 shows the simulation waveform of the no-load start process. As can be seen from Figure 17, the motor starts from a standstill, and then sets the reference speeds of 150 rpm, 300 rpm, and 100 rpm, and makes a smooth transition between different reference speeds. During the simulation process, the maximum rotor position error does not exceed 0.12 rad, the maximum speed error does not exceed 40 rpm, and the speed overshoot does not exceed 21 rpm.

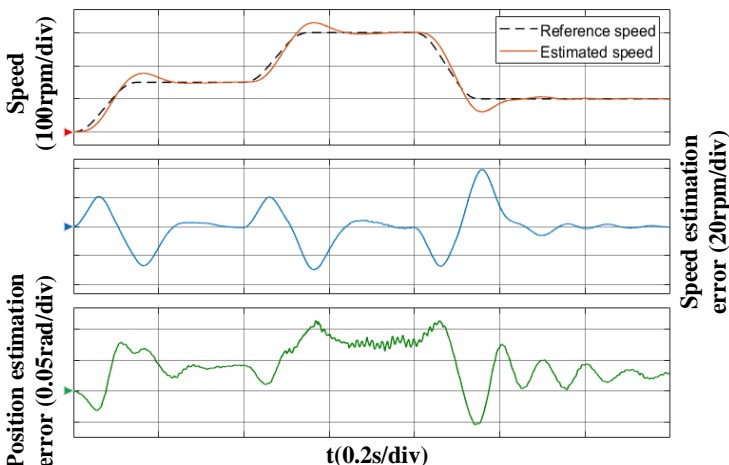

**Figure 17.** Simulation waveform of no-load starting process.

Figure 18 shows the simulation waveform of the motor running with load of 3 N.m. As can be seen from Figure 18, the motor starts from a standstill, and then sets the reference speeds of 150 rpm, 300 rpm, and 100 rpm, and makes a smooth transition between different reference speeds. During the simulation process, since the electromagnetic torque of the motor increases from 0, it will reverse due to the influence of the load torque, but it can quickly track the reference speed, so the system has a certain robustness, and the maximum rotor position error does not exceed 0.34 rad, the maximum speed error does not exceed 63 rpm, and the speed overshoot does not exceed 30 rpm. Therefore, the sensorless control system still has better dynamic performance under load.

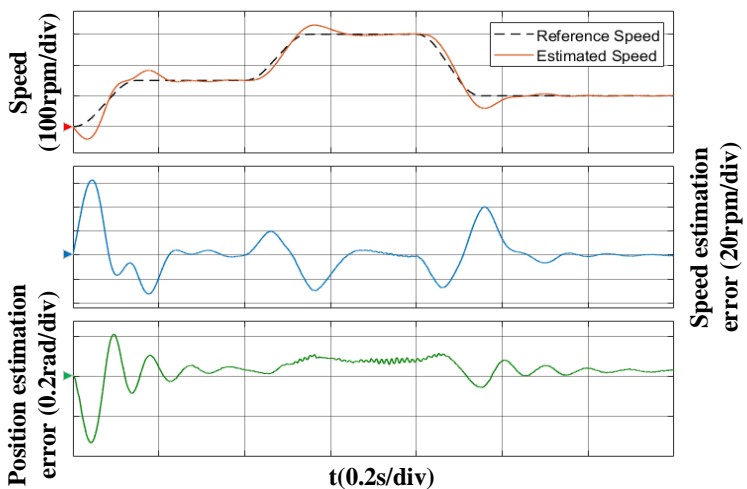

**Figure 18.** Simulation waveform of load starting process.

## 5. Conclusions

In this paper, the HF square wave voltage injection method is used to realize the sensorless control in the range of zero and low speed at 1 kHz switching frequency. At the low switching frequency, due to the limited injection frequency, the update frequency of the position quadrature signal extracted by the square wave voltage injection method is lower than the PWM update frequency, which leads to the problem that the estimated position ripple is too large.

A method based on CIC interpolation filtering is proposed in this paper. By this method, when the injected square wave voltage frequency is 500 Hz, the signal-processing strategy reduces the harmonic content of the position quadrature signal while increasing the sampling rate of the signal. Then, the delay angle is compensated by using the PWM

update frequency and the estimated rotational speed. Simulation results show that the position estimation error of the system does not exceed 0.34 rad, and the speed estimation error does not exceed 63 rpm when the speed is adjusted in the low-speed area of not more than 300 rpm. Thus, the proposed sensorless control strategy can improve the control performance of the system.

**Author Contributions:** Literature search, Graph production, Study design, Data collection, Data analysis, Data processing, Manuscript writing, Z.X.; Manuscript review, Data analysis, L.L.; Manuscript Review, Data processing, X.W. (Xiaolu Wang); Literature Search, Graph Production, X.W. (Xin Wang). All authors have read and agreed to the published version of the manuscript.

**Funding:** This research received no external funding.

**Institutional Review Board Statement:** Not applicable.

**Informed Consent Statement:** Not applicable.

**Data Availability Statement:** Exclude this statement.

**Conflicts of Interest:** The authors declare no conflict of interest.

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
