# Peer review of "A Sensorless Control Strategy for Permanent Magnet Synchronous Motor at Low Switching Frequencyâ€"

_electronics, doi:10.3390/electronics11131957_

Round 1
Reviewer 1 Report
The paper presents a method for estimating the position of a permanent magnet synchronous motor (PMSM) using injected excitation currents. The layout of the presented material needs to be significantly overhauled. In particular, the work needs to lay out more clearly that it is only pertaining the rotor position estimation, and also delineate other parts which the authors do not contribute towards.
1. The system model (1) is inadequately described: it is not clear what parameters p and Ψ_f are.
2. What is FOC in Figure 1 stand for?
3. While there is a lot of background and introduction for the extraction of the high frequency (HF) response at the end of Section 2, the other methods and control loop are suddenly introduced without motivation or background theory. This makes the contribution of the work unclear, as it is uncertain what is assumed to be designed, and what the actual contribution of the work is.
4. Figure 2 is also very confusing - it is unclear which parts of the diagram denote the HF signal injection, and what the other blocks mean (Lomborg? CIC? PLL?) Furthermore, the contribution of the presented work is unclear, as it is not immediately obvious which parts the authors are working on, and which parts are assumed to be designed beforehand.
5. In Section 3, several methods are presented, which build on each other. Please provide a design method/algorithm for the method proposed in this paper for clarity.
6. Does CIC stand for anything? Please define it before its first use.
7. The authors mention the use of a Romberg observer, which is spelt differently from the Lomborg observer in Figure 2. Do they mean a Luenberger observer? Please provide a citation to the full equations for the observer used. How is the observer designed to estimate the rotor position and mechanical angular velocity?
Author Response
Point 1: The system model (1) is inadequately described: it is not clear what parameters p and Ψ_f are.
Response 1: This is a typical mathematical model of a permanent magnet synchronous motor in a rotating coordinate system, where p is the symbol for partial derivatives; Ψf is the flux linkage. The meaning of these symbols has been added to the paper for better clarification.
Point 2: What is FOC in Figure 1 stand for?
Response 2: FOC refers to field-oriented control (FOC). It is a closed-loop control strategy of the motor mentioned in the paper. To avoid ambiguity, the ‘FOC period’ in Figure 1 is replaced by ‘closed-loop control period’ in this paper.
Point 3: While there is a lot of background and introduction for the extraction of the high frequency (HF) response at the end of Section 2, the other methods and control loop are suddenly introduced without motivation or background theory. This makes the contribution of the work unclear, as it is uncertain what is assumed to be designed, and what the actual contribution of the work is.
Response 3: The last paragraph of the second chapter is a part of the sensorless control strategy, which briefly describes the basic principles from extracting the current signal to estimating the rotor position, and provides a theoretical basis for the design and innovation of the subsequent sections. In order to avoid misunderstandings, the role of each step in the entire control strategy is briefly described, so that this part of the content can be better related to the subsequent sections.
Point 4: Figure 2 is also very confusing - it is unclear which parts of the diagram denote the HF signal injection, and what the other blocks mean (Lomborg? CIC? PLL?) Furthermore, the contribution of the presented work is unclear, as it is not immediately obvious which parts the authors are working on, and which parts are assumed to be designed beforehand.
Response 4: Figure 2 is the block diagram of PMSM control system using square wave voltage injection method proposed in this paper. It includes the classic field-oriented control (FOC) strategy in the field of motor control, as well as the proposed high-frequency injection related algorithm.
The full name of each blocks is supplemented in the paper, and the specific principle is shown in the following sections. Among them, CIC is the cascade integrator comb interpolation filtering, PLL is a phase-locked loop, and Luenberger observer is a reduced-order observer in modern control theory.
Point 5: In Section 3, several methods are presented, which build on each other. Please provide a design method/algorithm for the method proposed in this paper for clarity.
Response 5: In each subsection of section 3, the existing problems in other papers are described first, and then an improved method for this problem is proposed.
Point 6: Does CIC stand for anything? Please define it before its first use.
Response 6: CIC is the abbreviation of Cascade Integrator Comb, which has been firstly described in the abstract, and then pre-denfined in the last paragraph of section 2 in the main body texts.
Point 7: The authors mention the use of a Romberg observer, which is spelt differently from the Lomborg observer in Figure 2. Do they mean a Luenberger observer? Please provide a citation to the full equations for the observer used. How is the observer designed to estimate the rotor position and mechanical angular velocity?
Response 7: Thank you for spotting this spelling error, it has now corrected from ' Romberg' to ' Luenberger '. In addition, the observer structure, principle and stability analysis are added to this paper, with the corresponding citation (reference [26]).

Reviewer 2 Report
The work under review proposes a method for extracting position quadrature signals based on sampling rate transformation, and a signal processing strategy based on Cascade Integrator Comb interpolation filtering. The results are convincing, but the work needs to be revised considering the following concern:
1. The main motivation for your proposal seems not to be studied adequately.
2. The novelty of the advocated approach with respect to the previous studies in the relevant field must be emphasized. The contribution of the work is vague.
3. The stability and robustness analysis to varying parameters are to be exercised.
4. Please investigate the effect of frequency on the proposed approach.
5. In simulations, various working conditions must be used to test the performance of the proposed method. Given an example, use different speed references and load disturbances in order to acquire the position error dynamically.
6. It is strongly suggested that the performance of the introduced approach be verified experimentally.
7. Regarding the literature survey of PMSMs, the following papers are to be reviewed: (1) An educational tool for the genetic algorithm-based fuzzy logic controller of a permanent magnet synchronous motor drive (2) An adaptive PI controller schema based on fuzzy logic controller for speed control of permanent magnet synchronous motors.
Author Response
Point 1: The main motivation for your proposal seems not to be studied adequately.
Response 1: The motivation of the paper is stated in the abstract, which the main purpose is to address the problem of increased position estimation error due to signal distortion in sensorless systems at low switching frequency. In order to solve this problem, this paper suggests a series of improvement strategies, such as position quadrature signal extraction, CIC filtering, delay compensation, etc., to improve the signal-to-noise ratio and reduce delay, and avoid the adverse effects of low switching frequency on the control system. Finally, the switching frequency of 1kHz is selected in the simulation. It should be noted that the common switching frequency of motor control is 10k~20kHz, and the low switching frequency should not be greater than 1kHz.
Point 2: The novelty of the advocated approach with respect to the previous studies in the relevant field must be emphasized. The contribution of the work is vague.
Response 2: The main innovation of this paper is reflected and elaborated in the third section by analyzing and comparing the existing methods of extracting high-frequency signals in the paper [19-21], and innovatively proposes the method for extracting position quadrature signals. Then, the method of extracting envelopes in the articles [22-23] is also analyzed and compared, followed by a novel method of extracting envelopes is proposed. In addition, after analyzing the interpolation filtering method proposed in the paper [24], a signal improvement method based on CIC filtering is proposed. All of the above methods are all original methods that contribute to this article, and they have certain advantages over other articles in the specific case of low switching frequency.
Point 3: The stability and robustness analysis to varying parameters are to be exercised.
Response 3: The stability analysis of the system and the selection range of the Luenberger observer parameters are analyzed.
Point 4: Please investigate the effect of frequency on the proposed approach.
Response 4: There are many articles studying the effect of low switching frequency on control performance, not only including the article referenced in this paper [10]. At low switching frequency, the high-frequency response current is distorted seriously, and the position estimation error increases. This has become a consensus among scholars. Therefore, instead of repeating the existing findings, this paper provides the following reference papers for description. Among them, the optimization methods under low switching frequency include not only the improvement of the signal extraction method, but also the active compensation of the delay [a], and the improvement of the modulation method [b].
[a] Y. Wang, Z. Xue, G. Luo and Z. Chen, "A Time-Delay Compensation Method for PMSM Sensorless Control System under Low Switching Frequency," IECON 2019 - 45th Annual Conference of the IEEE Industrial Electronics Society, 2019, pp. 880-885, doi: 10.1109/IECON.2019.8927321.
[b] H. Zhang, W. Liu, Z. Chen, G. Luo, J. Liu and D. Zhao, "Asymmetric Space Vector Modulation for PMSM Sensorless Drives Based on Square-Wave Voltage-Injection Method," in IEEE Transactions on Industry Applications, vol. 54, no. 2, pp. 1425-1436, March-April 2018, doi: 10.1109/TIA.2017.2772166.
Point 5: In simulations, various working conditions must be used to test the performance of the proposed method. Given an example, use different speed references and load disturbances in order to acquire the position error dynamically.
Response 5: The simulation of motor speed regulation after no-load starting and the simulation of motor speed regulation after on load starting are supplemented. The simulation results show that the system has good dynamic performance and load carrying capacity.
Point 6: It is strongly suggested that the performance of the introduced approach be verified experimentally.
Response 6: After analysis, more details of simulation and implementation are added in the paper, and more simulation results are added to improve confidence. This is a more efficient way to verify in my option.
Point 7: Regarding the literature survey of PMSMs, the following papers are to be reviewed: (1) An educational tool for the genetic algorithm-based fuzzy logic controller of a permanent magnet synchronous motor drive (2) An adaptive PI controller schema based on fuzzy logic controller for speed control of permanent magnet synchronous motors.
Response 7: By reading the above two references, I think these two papers have certain reference significance for this paper, so I quoted the papers.

Reviewer 3 Report
The paper has proposed a sensorless control strategy for PMSMs based on the HF square wave voltage injection method and CIC interpolation filtering. My comments can be found below:
1) Please supplement the meaning of employed abbreviations in the main body.
2) In Line 68 - it is not rigorous to state that a method is varied by simulation as the simulation itself needs to be verified by experiments! Therefore, it is also recommended to modify the title of Section 4 to something like "Case study".
3) It is hard to capture the logic of the paragraph in Lines 74-79: why do the authors mention that the inductance is greater than the resistance in a winding but then the voltage drop due to the inductance can be neglected? Please give more explanations.
4) Please cross-check the typo and grammar issues of the whole paper. For example, I assume that the letter "d" in "d-axis" should be italic, e.g., in Line 84? In line 95 the word "filter" is countable. In Line 124 please add "the" before "PMSM control system"...
5) It is not appropriate to use "where" below a figure to introduce the content, e.g., Figures 3, 4, and 8. It is better to use "In figure 3..."
6) Line 154 "Th" without a subscription.
7) Line 176 "in the figure" is not clear... you mean Figure 5?
8) In Figures 6 and 9 why not use legends to characterize the curves?
9) In Line 204, I am not sure whether "frequency match" is proper terminology, I mean the authors may want to say "to match the frequencies of two signals?"
10) Again, in Line 206, "in the figure" is not clear...
11) Line 216 is confusing: the number is a multiplier of the rate increase or?
12) In Line 279 please do not use "tested" as experimental measurements are lacking in this paper... you mean "studied/investigated"?
13) Please do not use "something is good" like in Lines 283 and 294... the word "good" is not a rigorous word to describe the performance; please quantitatively describe the effectiveness.
14) In Section 5 please give more quantified conclusions rather than just qualitatively repeating the abstract/introduction.
Author Response
Point 1: Please supplement the meaning of employed abbreviations in the main body.
Response 1: Abbreviations and their expansions have been added to the paper to ensure that all terminologies are explained the first time they appear.
Point 2: In Line 68 - it is not rigorous to state that a method is varied by simulation as the simulation itself needs to be verified by experiments! Therefore, it is also recommended to modify the title of Section 4 to something like "Case study".
Response 2: Firstly, as a verification method, simulation has the problem of insufficient verification, which is a shortcoming of this paper. In order to further improve the reliability of the simulation results, more design processes are added in section 3, and more simulation details are added in section 4. Finally, the title of section 4 has been changed to 'Simulation Analysis' to enhance accuracy.
Point 3:It is hard to capture the logic of the paragraph in Lines 74-79: why do the authors mention that the inductance is greater than the resistance in a winding but then the voltage drop due to the inductance can be neglected? Please give more explanations.
Response 3: Firstly, the motor consists of resistive and inductive loads, but in the case of high-frequency voltage injection, due to the sudden change of current, the voltage drop across the inductor will be much larger than the voltage drop generated by the inductor itself and the voltage drop across the stator resistance. Therefore, the voltage drop of the inductor itself and the voltage drop of the stator resistance, as well as the back EMF are ignored in the paper.
Point 4: Please cross-check the typo and grammar issues of the whole paper. For example, I assume that the letter "d" in "d-axis" should be italic, e.g., in Line 84? In line 95 the word "filter" is countable. In Line 124 please add "the" before "PMSM control system"...
Response 4: After careful inspection, formatting errors in the paper have been corrected.
Point 5:It is not appropriate to use "where" below a figure to introduce the content, e.g., Figures 3, 4, and 8. It is better to use "In figure 3..."
Response 5: This is a good suggestion, I have changed the way of expression to introduce the content directly.
Point 6:Line 154 "Th" without a subscription.
Response 6: After equation 3, added the supplement for "Th". "Th" is the injection period of the HF square wave voltage. Due to the continuous-pulse injection method. Thus, the value of "Th" equal to half the control period.
Point 7:Line 176 "in the figure" is not clear... you mean Figure 5?
Response 7: This is a similar question to point5. The description in the text has been changed to ‘As shown in the figure 5’.
Point 8:In Figures 6 and 9 why not use legends to characterize the curves?
Response 8: Using legend may obscure some curves and take up more space in figures. After comparison, the former is adopted.
Point 9:In Line 204, I am not sure whether "frequency match" is proper terminology, I mean the authors may want to say "to match the frequencies of two signals?"
Response 9: Here, its meaning is matching the frequency of two signals. However, the description in the text is not very clear, and has been modified.
Point 10:Again, in Line 206, "in the figure" is not clear...
Response 10: The description in the text has been changed to ‘As shown in the figure 7’.
Point 11:Line 216 is confusing: the number is a multiplier of the rate increase or?
Response 11: The description is not clear, it means that the interpolation points increase by the same factor as the sampling rate. This has been modified in the paper.
Point 12: In Line 279 please do not use "tested" as experimental measurements are lacking in this paper... you mean "studied/investigated"?
Response 12: It means that the proposed method or strategy has been tested by simulation. This has been supplemented in the paper.
Point 13: Please do not use "something is good" like in Lines 283 and 294... the word "good" is not a rigorous word to describe the performance; please quantitatively describe the effectiveness.
Response 13: A number of quantitative evaluations of the simulation results have been added to the paper.
Point 14: In Section 5 please give more quantified conclusions rather than just qualitatively repeating the abstract/introduction.
Response 14: Several quantitative evaluation results have been added to the conclusion of the paper to improve confidence.

Round 2
Reviewer 1 Report
Many of my comments have not been addressed satisfactorily, if at all.
1. Regarding my previous comment 1, please cite the existing model and also explain the parameters better. In particular, it is not clear what is meant by `partial derivatives'.
2. Regarding my previous comment 3, the amendment made was to list down the methods used - this still does not motivate the methods (which should be done in the Introduction), and how they compare over existing methods. Essentially, no justification for the methods are made.
3. Regarding my previous comment 4, the Figure is still unclear - it is not immediately obvious which part belongs to the control loop, which portion extracts the current, and so on.
4. Regarding my previous comment 5, no elaboration was provided at all.
5. Regarding my previous comment 7, the design of the observer is still unclear. The equations for the observer are not provided, thus it is unclear how the analysis carried out in subsection 3.3 was performed.
6. Also check the numbering of sections, particularly for the Results.
Author Response
Point 1: Regarding my previous comment 1, please cite the existing model and also explain the parameters better. In particular, it is not clear what is meant by `partial derivatives'.
Response 1: I am so sorry, the previous description of the motor model in this section may not be clear. After referring to and citing relevant paper, the model is explained as follows:
For permanent magnet synchronous motor, the d-q axis model in two-phase rotating coordinate system is relatively simple, and the inductance of the motor is independent of the rotor position. The coordinate system takes the rotor as the reference system, and the difference between the two shafts is 90 °. It is similar to direct-current (DC) motor, d-axis is equivalent to excitation winding, and q-axis is equivalent to armature winding [a]. Therefore, the position estimation strategy proposed in this paper is based on the motor equation in the rotating coordinate system. The expanded form of the voltage equation of the motor in the rotating coordinate system is:
Equivalent transformation of the above formula into the form of matrix, and the derivative operation is expressed by p, that is, p=d/dt. Then the following equation can be obtained [b-c]:
where p is differential operator. To sum up, here is just a change in the form of the equation, and the symbol p is used to represent d/dt. These questiones has been revised in the paper.
[a] Salminen P, Pyrhönen J, Niemelä M. A Comparison Between Surface Magnets and Embedded Magnetsin Fractional Slot Wound PM Motors[C]// Isef 2003 -, International Symposium on Electromagnetic Fields in Electrical Engineering Maribor, Slovenia, September. 2005:209-214.
[b] H. Zhang, W. Liu, Z. Chen, G. Luo, J. Liu and D. Zhao, "Asymmetric Space Vector Modulation for PMSM Sensorless Drives Based on Square-Wave Voltage-Injection Method," in IEEE Transactions on Industry Applications, vol. 54, no. 2, pp. 1425-1436.DOI: http://doi.org/10.1109/TIA.2017.2772166
[c] Y. Lee, Y. Kwon and S. Sul, "Comparison of rotor position estimation performance in fundamental-model-based sensorless control of PMSM," 2015 IEEE Energy Conversion Congress and Exposition (ECCE), 2015, pp. 5624-5633, doi: 10.1109/ECCE.2015.7310451.
Point 2: Regarding my previous comment 3, the amendment made was to list down the methods used - this still does not motivate the methods (which should be done in the Introduction), and how they compare over existing methods. Essentially, no justification for the methods are made.
Response 2: There was a bias in the previous understanding of the comment 3, which resulted in an inaccurate reply. The specific explanation is as follows:
Firstly, the position sensorless strategy based on voltage square wave injection adopted in this paper is introduced in detail in Section 2, and in the last paragraph of Section 2, some improved methods based on this injection strategy are outlined. This makes it relevant to the last paragraph in the Introduction on the motivation for proposing these strategies, and also facilitates subsequent sections such as strategy design and validation results. Therefore, in the first revision, the second part was supplemented.
In addition, the reasons for proposing these methods are explained in the Introduction. Specifically as follows:
- By referring to the paper [1-2], the advantages of the permanent magnet synchronous motor are compared and analyzed, and the permanent magnet synchronous motor is finally determined as the controlled object.
- By referring to the paper [3-11], different position sensorless methods are compared, and the high-frequency square wave voltage injection method is selected in combination with the working conditions mentioned in the article [12].
- Since the high-frequency square-wave voltage injection method has current harmonics and high-frequency noise [13], and the neglected resistance at low speed affects the position estimation accuracy [14-16], it is necessary to apply the traditional high-frequency square wave Improved voltage injection method.
- At present, articles [17-20] propose some improvement methods, but they all require higher switching frequency so that one or more injected HF signals can be inserted in multiple carrier cycles, which is not suitable for low The case of switching frequency.
- In order to solve these problems and improve the position observation accuracy and control performance at low switching frequency, this paper bases on the high frequency square wave voltage injection method, proposes a method for extracting position quadrature signals based on sampling rate transformation to avoids the use of complex modulation methods, and this paper adopts the signal processing strategy based on the CIC interpolation filtering an integral cascade comb interpolation filter is proposed to solve the problem of mismatch between the update frequency of the position quadrature signal and the PWM update frequency, and suppress the HF ripple caused by the low update frequency of the position quadrature signal. At the same time, the delay compensation strategy is used to compensate for the delay caused by position estimation and inverter update, which effectively improves the accuracy of position estimation.
The above are the background and motivation of this study. In order to avoid unclear understanding, the Introduction section is supplemented and explained in the paper.
Point 3: Regarding my previous comment 4, the Figure is still unclear - it is not immediately obvious which part belongs to the control loop, which portion extracts the current, and so on.
Response 3: Since the closed-loop control part and the position estimation part are not marked in the previous figure, the understanding is troubled. In response to this problem, Figure 2 has been modified to distinguish the FOC-based closed-loop control part and the part by extracting high-frequency currents and estimating rotor position and speed with different color blocks.
Point 4: Regarding my previous comment 5, no elaboration was provided at all.
Response 4: Each subsection in the third section corresponds to the design method proposed in this paper, as shown in the following table.
Section |
The strategy proposed in this paper |
Implementation |
Innovation |
3.1 |
(1) High Frequency Response Current Extraction Method |
(1) Based on the delay module. |
(1) It solves the difficult problem of HF signal extraction, and avoids the use of various filters as in the paper [23-25]. |
(2) High Frequency Response Current Envelope Extraction Method |
(2) Based on the phase current sampling rate transformation module. |
(2) A simpler method is used to extract the envelope, compared with the paper [26-27]. |
|
3.2 |
Frequency Matching Method |
Based on single-order CIC interpolation filtering. |
When the switching frequency is more than twice the sampling frequency, the quadratic interpolation can only double the sampling points, which still cannot effectively solve the waveform distortion caused by the sampling holding effect [28]. Based on the CIC theory in [29], this paper applies it to a position sensorless system, to improve the signal-to-noise ratio of the signal. |
Finally, in order to facilitate understanding, the structures in Section 3 have been modified, The basic structure is: (1) the current method in others paper, (2) Analyze and compare problems in others papers (3) The improved method proposed and design process in this paper.
Point 5: Regarding my previous comment 7, the design of the observer is still unclear. The equations for the observer are not provided, thus it is unclear how the analysis carried out in subsection 3.3 was performed.
Response 5: I am very sorry that my previous description confused the reviewer. After referring to relevant paper [d-f], the question is explained as follows:
In this paper, the principle of the Luenberger observer is shown through the block diagram of the structure of the observer, and the transfer function of the observer based on the block diagram is listed (see Equation 9).
Then, the analysis is carried out according to Equation 9 of the observer and Equation 10 of the motor of motion. I thought before that the design details of the observer can be explained by the structure block diagram of the observer.
In order to increase the credibility of the observer design process, this paper supplements the observer equation and references paper, see [30-31] for details.
[d] Q. Lu, Y. Wang, L. Mo and T. Zhang, "Pulsating High Frequency Voltage Injection Strategy for Sensorless Permanent Magnet Synchronous Motor Drives," in IEEE Transactions on Applied Superconductivity, vol. 31, no. 8, pp. 1-4, Nov. 2021, Art no. 5204204, doi: 10.1109/TASC.2021.3094426.
[e] H. Zhang, W. Liu, Z. Chen, G. Luo, J. Liu and D. Zhao, "Asymmetric Space Vector Modulation for PMSM Sensorless Drives Based on Square-Wave Voltage-Injection Method," in IEEE Transactions on Industry Applications, vol. 54, no. 2, pp. 1425-1436.DOI: http://doi.org/10.1109/TIA.2017.2772166.
[f] Y. Zhang, Z. Yin, J. Liu, R. Zhang and X. Sun, "IPMSM Sensorless Control Using High-Frequency Voltage Injection Method With Random Switching Frequency for Audible Noise Improvement," in IEEE Transactions on Industrial Electronics, vol. 67, no. 7, pp. 6019-6030, July 2020, doi: 10.1109/TIE.2019.2937042.
Point 6: Also check the numbering of sections, particularly for the Results.
Response 6: The number of each chapter and figure in the paper has been checked, and the problem items have been revised to avoid the inconsistency of each number.

Reviewer 2 Report
The revised version of the article addresses the concerns of this reviewer and may be accepted in this journal.
Author Response
Thank you very much for your comments on this paper.
Reviewer 3 Report
The current version looks acceptable to me.
Author Response

(The authors gave the same response as above.)
